# Separating signal from noise: a self-distillation approach for amortized heterogeneous cryo-EM reconstruction

## Abstract

Cryogenic electron microscopy (cryo-EM) has become an essential tool in structural biology for determining dynamic biomolecular structures at high resolution. However, state-of-the-art VAE-based reconstruction methods such as CryoDRGN do not generalize to new particle images: its encoder overfits to the training data due to the presence of high amounts of noise. In this work, we propose a simple yet effective strategy to generalize to images not in the training set: learning noise invariant representations. We propose **Cryo-No-O**verfit (CryoNOO), which extends CryoDRGN via self-distillation by leveraging the reconstruction method itself as a denoiser to generate augmented views of training images. We then learn noise-invariant representations via self-supervised learning, enabling reconstruction methods to amortize inference to unseen images. Extensive empirical evaluations on both synthetic and experimental datasets demonstrate that our method dramatically improves reconstruction quality on unseen test data, marking a key step towards robust, generalizable cryo-EM reconstruction.

## 1 Introduction

Single-particle cryo-electron microscopy (cryo-EM) has revolutionized structural biology by enabling the determination of macromolecular complex structures at near-atomic resolution (Nakane et al., 2018; Yip et al., 2020). Unlike structure prediction tools such as AlphaFold (Jumper et al., 2021; Abramson et al., 2024), which predict a static model from amino acid sequences, cryo-EM has the distinct advantage of capturing dynamic conformational ensembles of molecules under experimental conditions. Understanding the conformational ensemble of molecules is essential for understanding biological processes such as allosteric regulation, enzyme catalysis, and ligand binding.

However, heterogeneous reconstruction from cryo-EM data presents a challenging inverse problem, as each 2D image is a noisy projection of a single 3D conformation in an unknown orientation; the task is further complicated by conformational heterogeneity and the extremely noisy images produced by low-dose acquisition required to minimize radiation damage. These characteristics lead to a fundamentally ill-posed inverse problem with incomplete and noisy observations. A series of deep-learning based methods tackle the problem of cryo-EM heterogeneous reconstruction (Donnat et al., 2022). These methods generally follow the structure of an autoencoder, where an MLP- or CNN-based encoder maps a 2D cryo-EM image to a latent vector that is then decoded into a 3D volume and rendered to obtain supervision.

One potential advantage of autoencoding is *amortized inference*, or the ability to quickly perform reconstruction of a new, unseen cryo-EM image with a single forward pass by feeding it to the (pre-)trained encoder. However, it is common knowledge that the encoder of these approaches overfits to the training images and fails to generalize, and we refer to Edelberg & Lederman (2023) for a detailed study. Even small amounts of noise can cause an encoder to overfit, and while it can reconstruct high resolution structures from training images, it fails to learn meaningful representations on unseen images of the same structure. This observation highlights a key gap in the field: There are no existing methods that explicitly target generalizability under realistic, noisy conditions.

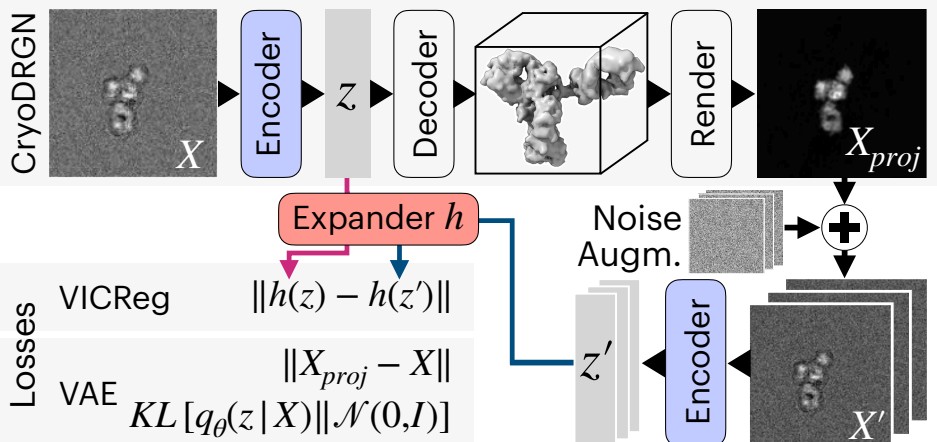

Figure 1: **Method Overview.** A training image $X$ is first fed through the CryoDRGN VAE, producing a latent $z$ as well as a reconstruction of the input image $X_{proj}$ rendered from the reconstructed 3D structure. Our method augments this reconstruction with additive Gaussian noise, generating a set of augmentations $X'$. We then pass these through the *same* encoder to produce latents $z'$, and ask that these latents match the latent produced by the original training image $z$. To prevent latent collapse, we leverage the VICReg framework, which expands $z$ and $z'$ via an expander MLP $h$ before penalizing their least-squares differences. The model is trained end-to-end with the original CryoDRGN VAE loss and an additional VICReg loss.

In this paper, we introduce Cryo-NO-Overfit (CryoNOO), a training paradigm for autoencoder-based cryo-EM heterogeneous reconstruction methods that enables the model to generalize to unseen cryo-EM images at inference time for the first time. Our key insight is to use the autoencoder architecture itself as both a denoising mechanism and a generator of novel views of protein structures in the training set, enabling a form of self-distillation. Specifically, we reconstruct a 3D structure from a noisy training image and reproject it into 2D to synthesize clean views. These views can then be augmented with controllable noise levels to generate diverse inputs for self-supervised training.

We then enforce consistency in the latent space by requiring the encoder to map both the original and augmented views to the same latent code. To prevent trivial solutions, we leverage Variance-Invariance-Covariance Regularization (VICReg) Bardes et al. (2021) which stabilizes training by preserving variance and decorrelation while reducing the norm of the difference of latent encodings.

In summary, our work makes the following key contributions:

- We propose learning noise-invariant representations as way to generalize under noisy real world conditions.

- We introduce a self-distillation technique that leverages the encoder-based reconstruction method itself to generate augmented views of training samples and use self-supervised learning to enforce invariance.

- We empirically validate our method across synthetic and real cryo-EM datasets and, for the first time, demonstrate amortized heterogeneous 3D reconstruction of novel particles unseen during training.

## 2 RELATED WORK

**Cryo-EM Reconstruction Methods.** Neural network-based cryo-EM reconstruction methods commonly employ VAE architectures to model continuous structural heterogeneity via amortized inference. CryoDRGN (Zhong et al., 2021), for example, learns a function $\hat{V} : \mathbb{R}^{3+n} \to \mathbb{R}$ that maps a low-dimensional latent space to a 3D electron scattering potential in the Fourier domain. One line of research in this area is concerned with amortizing inference in order to speed up reconstruction (Levy et al., 2022a;b; 2024). In the *ab initio* setting, CryoAI (Levy et al., 2022a) and CryoFire (Levy et al.,

2022b) proposed amortized inference of poses by jointly predicting the pose. Other works investigate the amortized inference of the latent conformation. As observed by previous works (Edelberg & Lederman, 2023; Levy et al., 2024), cryoDRGN fails to generalize to unseen views and tends to memorize the training images.

CryoDRGN-AI (Levy et al., 2024) circumvents possible memorization by dispensing with the encoder altogether, instead using an autodecoder architecture. Unlike autoencoders, where an encoder maps 2D views to latent codes, autodecoders do not have an encoder and instead give each view a unique learnable latent code, stored in a lookup table. To perform inference on views unseen during training, a new latent code is created and optimized during test time, making inference more expensive than autoencoding, which *amortized* inference, only requires a forward pass of its encoder. In contrast, we propose to perform amortized inference in the variational autoencoder framework by learning a noise-invariant encoder with self-distillation.

**Self-Supervised Learning and Self-Distillation.** In self-supervised learning from images, we are interested in learning an encoder that produces latent representations useful for downstream tasks using only unlabeled images. This is usually accomplished by encoding augmentations of the same image and enforcing them to be similar in latent space. The core challenge is to avoid trivial solutions, such as predicting a constant representation independent of the encoded image. This can be accomplished in different ways. In this paper, we rely on VICReg (Bardes et al., 2021) which avoids representational collapse by enforcing three principles: encouraging consistency across views, maintaining diversity across samples, and reducing redundancy across feature dimensions. Also related to our method are methods of self-distillation, where a "student" encoder is distilled from a "teacher" whose weights are however an exponential moving average of the student's weights (Grill et al., 2020; Chen & He, 2021; Caron et al., 2021). Our method can be seen as a form of self-distillation, where we are distilling knowledge already inherent in the CryoDRGN model into a more powerful encoder, though we rely on VICReg to prevent representational collapse. VICReg has no notion of generating pseudo-groundtruth with a 3D reconstruction model, and to the best of our knowledge, VICReg has never been used in combination with a differentiable forward model or differentiable renderer.

# 3 AMORTIZED HETEROGENEOUS CRYO-EM RECONSTRUCTION

## 3.1 PRELIMINARIES

**Cryo-EM Image Formation Model.** Cryo-electron microscopy (cryo-EM) aims to reconstruct a 3D molecular structure $V : \mathbb{R}^3 \to \mathbb{R}$ from noisy 2D projection images $X_1, \ldots, X_N$, each corresponding to a particle in an unknown pose defined by a rotation $R \in SO(3)$ and an in-plane translation $t \in \mathbb{R}^2$. The image formation model can be written as

$$X_i = C_i \, P_{\phi_i} V + \eta_i, \tag{1}$$

where $C_i$ models the contrast transfer function (CTF), $P_{\phi_i}$ denotes the projection operator for pose $\phi_i = (R_i, t_i)$, and $\eta_i \sim \mathcal{N}(0, \sigma^2)$ is additive noise. In practice, reconstruction is typically performed in Fourier space using the Fourier Slice Theorem (FST) Bracewell (1956), which shows that each 2D image corresponds to a central slice of the 3D Fourier transform of $V$.

**CryoDRGN as an implicit teacher.** CryoDRGN models structural heterogeneity with a VAE where the encoder $q_\xi(\mathbf{z}|X)$ with parameters $\xi$ maps each particle image to a latent vector $\mathbf{z}$ representing structural variability. This latent representation is passed to a conditional implicit neural representation (INR) decoder to reconstruct a 3D structure, which is then projected into the 2D image domain via the cryo-EM image formation model (Sec. 3.1) and compared to the input image using a standard VAE loss. Although trained with a variational objective, CryoDRGN effectively *denoises*: it reconstructs a clean 3D structure from noisy 2D inputs. This observation enables us to view the model itself as a *teacher* that can supervise later iterations of its own encoder – a form of *self-distillation*.

**VICReg.** To prevent representation collapse in self-supervised learning, VICReg (Bardes et al., 2021) introduced a loss composed of three terms: invariance, variance and covariance regularization. These operate on the representation space produced by an extra expander network $h$ to enforce non-triviality and disentanglement across embedding dimensions.

The *variance* term encourages each embedding dimension to exhibit sufficient variation across the batch using a hinge function, while the *covariance* term penalizes correlations between dimensions:

$$v(z) = \frac{1}{d} \sum_{j=1}^{d} \max\left(0, \gamma - \sqrt{\text{Varbatch}(h(z_j)) + \epsilon}\right), \quad c(z) = \frac{1}{d} \sum i \neq j \left[\frac{1}{n-1} h(Z)^\top h(Z)\right]_{i,j}^2, \tag{2}$$

where $\gamma = 1$ is the target standard deviation, $\epsilon$ is a small constant for numerical stability, and $Z \in \mathbb{R}^{n \times d}$ is the batch of centered embeddings. The *invariance loss* enforces the consistency between embeddings $h(z)$ and $h(z')$ of different augmentations of the same input:

$$s(z, z') = \frac{1}{n} \sum_{i=1}^{n} \|h(z_i) - h(z_i')\|_2^2. \tag{3}$$

The total VICReg loss combines these terms:

$$\mathcal{L}_{\text{VICReg}}(z, z') = \lambda_{\text{recon}} s(z, z') + \lambda_{\text{var}}(v(z) + v(z')) + \lambda_{\text{cov}}(c(z) + c(z')) \tag{4}$$

where $\lambda_{\text{recon}} = \lambda_{\text{var}} = \lambda_{\text{cov}} = 1$ by default. VICReg has primarily been used for image classification, object detection, and retrieval. To the best of our knowledge, this is the first instance of self-supervised learning techniques applied to cryo-EM reconstruction to enable generalization to new data

### 3.2 CryoNOO

In Cryo-No-Overfit (CryoNOO), we propose a new self-supervised learning objective that enables heterogeneous cryo-EM reconstruction methods to reconstruct images not in the training set. While our method can in principle work with any autoencoder-based model, we use the variational autoencoder (VAE) architecture of CryoDRGN. Our method can be viewed as a simple extension that enables the encoder to separate signal from noise.

**Definitions.** We define a CryoDRGN model as an encoder $q_\xi(\mathbf{z}|X)$ that takes an image $X$ and produces a latent $\mathbf{z}$ along with a decoder $p_\theta(z, k)$ that takes a latent $\mathbf{z}$ and coordinate $k = (k_x, k_y, k_z)$ in Fourier space and produces the Fourier transform of the electron scattering at $k$. $\xi$ and $\theta$ are the weights of the encoder and decoder, respectively. In cryo-EM reconstruction, we are given a noisy input image $I = T(X), T \sim \mathcal{T}$, the set of transformations that apply additive Gaussian noise $\mathcal{N}(0, \sigma)$ in real space, where $T$ and $X$ are unknown (Eqn. 6).

**Denoising and Self-Distillation.** Our key insight is that CryoDRGN inherently acts as a denoiser, as it produces clean 3D reconstructions from noisy input images. We leverage this denoising property to use CryoDRGN as its own teacher, performing self-distillation in the form of generating noise-augmented views for self-supervised learning. In order to ensure that CryoDRGN is a strong denoiser, we first pre-train the model for $N$ epochs with the usual VAE objective. Then, in subsequent epochs, we perform denoising as follows: given a noisy input image $I = T_0(X), T_0 \sim T$, we compute its latent representation $\mathbf{z}_0 = q_\xi(T_0(X))$, reconstruct a denoised 3D volume using decoder $p_\theta$, and render a 2D image $\hat{X}$ using the cryo-EM image model (see Eqn. 6 and Fig. 1). Since $\hat{X}$ is in Fourier space, we first apply the inverse Hartley transform to convert $\hat{x}$ to a real-space image, add Gaussian noise $\epsilon \sim N(0, f_j \sigma)$, where noise factors $f_j, j = \{1, \ldots, J\}$ are hyperparameters, and apply the forward Hartley transform to obtain augmented views $T_1(X), \ldots, T_J(X)$. The noise level $\sigma$ is estimated from $X$ using a heuristic. Unless stated otherwise, $J = 3$, but even with a single augmented view ($J = 1$) we get competitive performance, as shown in Table 4.

**Self-supervised learning for noise-invariant representations.** With multiple augmented views $T(X) = T_0(X), \ldots, T_J(X)$, we perform self-supervised learning to encourage robust and noise-invariant representations. Each augmented image is passed through the encoder $q_\xi$ to obtain latent vectors $\mathbf{z}_1, \ldots, \mathbf{z}_J$. To enforce invariance across these noise-augmented views while preserving meaningful structural variation, we apply the VICReg (Bardes et al., 2021) loss, chosen for both its empirical effectiveness and architectural simplicity. Unlike contrastive approaches, VICReg does not require negative samples or separate encoders for different views. In addition to enforcing invariance, VICReg promotes decorrelation across embedding dimensions and enforces sufficient

Table 1: **Quantitative Performance on synthetic datasets.** Comparison between CryoDRGN and CryoNOO on train and test splits across three datasets. CryoNOO maintains performance on the train set while performing better on test views.

| Method | Split | IgG-1D | | IgG-1D (SNR 0.01) | | IgG-RL | |
|---|---|---|---|---|---|---|---|
| | | Mean (std) | Median | Mean (std) | Median | Mean (std) | Median |
| CryoDRGN | Train | 0.383 (0.015) | 0.387 | 0.352 (0.011) | 0.354 | 0.367 (0.020) | 0.369 |
| **CryoNOO** | | 0.383 (0.002) | 0.383 | 0.359 (0.005) | 0.359 | 0.361 (0.020) | 0.363 |
| CryoDRGN | Test | 0.330 (0.012) | 0.326 | 0.300 (0.021) | 0.297 | 0.316 (0.012) | 0.316 |
| **CryoNOO** | | **0.367** (0.015) | **0.372** | **0.334** (0.019) | **0.338** | **0.339** (0.014) | **0.340** |

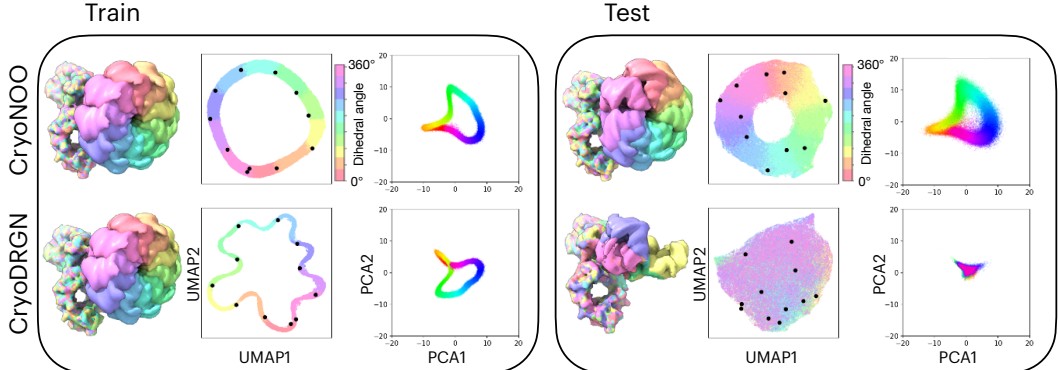

Figure 2: **Qualitative results for IgG-1D.** We find that CryoNOO can recover the 1D circular motion of the dataset in the test views, whereas CryoDRGN cannot recover this structure on the test dataset. This is also reflected in the latent spaces, where CryoNOO recovers the circular motion.

per-dimension variance, which helps prevent representation collapse and supports disentanglement of conformational variability. We note that conventional data augmentation in real space performs poorly (Table 7).

To use VICReg, we simply add a projection head $h$ (expander), implemented as a multi-layer perceptron (MLP), which maps each latent vector $\mathbf{z}_j$ to a higher-dimensional embedding space. The VICReg objective (see Section 3.1) is jointly optimized with the original VAE loss during training. The overall training objective is:

$$\mathcal{L}(X, \mathbf{z}_0, \ldots, \mathbf{z}_J) = \mathcal{L}_{\text{VAE}}(X, \mathbf{z}_0) + \lambda_{\text{VICReg}} \left( \sum_{j=1}^{J} \mathcal{L}_{\text{VICReg}}(\mathbf{z}_0, \mathbf{z}_j) \right) \quad (5)$$

where $\lambda_{\text{VICReg}}$ is a hyperparameter that controls the strength of the VICReg loss. We use 0.3 for this $\lambda_{\text{VICReg}}$. As in VICReg, the expander $h$ is only used for training and is thrown away after training, so after training we are left with a CryoDRGN model and inference proceeds identically to that of CryoDRGN. An overview of the full procedure is shown in Figure 1.

**Qualitative evaluation.** We evaluate latent spaces visually by mapping $Z_{\text{train}}, Z_{\text{test}}$ into a shared low dimensional space using a dimensionality reduction technique such as PCA or UMAP (McInnes et al., 2018). Unless otherwise stated, we visualize train and test distributions for the same model with a shared dimensionality reduction transformation.

## 4 RESULTS

We conduct extensive evaluation of CryoNOO on both synthetic and real datasets, presenting both qualitative and quantitative results for heterogeneous reconstruction and latent space consistency between training and test data distributions.

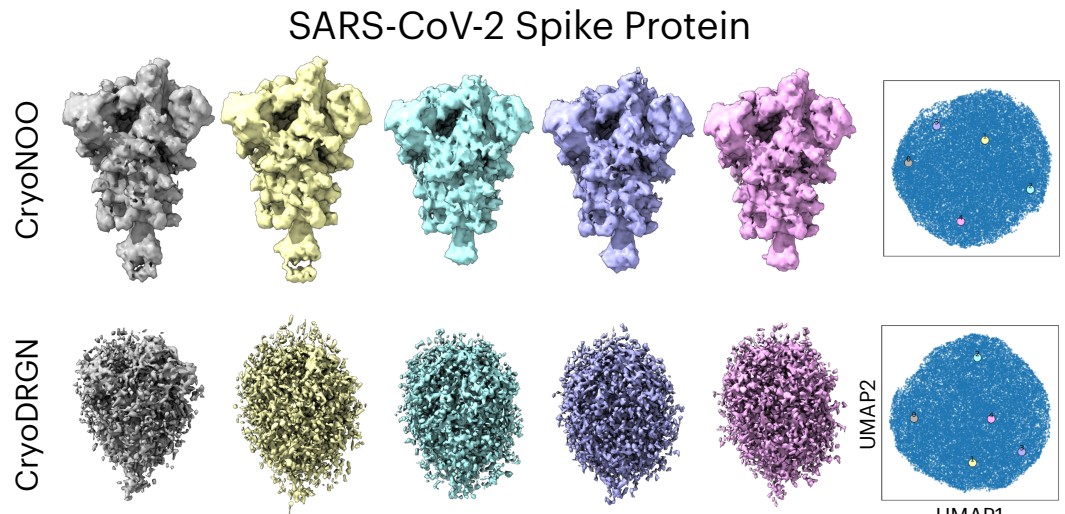

Figure 3: **SARS-CoV-2 Spike Protein.** Sampled density maps and latent embeddings visualized using UMAP, colored by K-means cluster labels ($k = 5$). Cluster centroids are indicated, and corresponding density maps illustrate centroid structures. CryoNOO is able to recover the structure of the protein on the test views, whereas CryoDRGN completely fails.

### 4.1 HETEROGENEOUS RECONSTRUCTION OF SYNTHETIC DATASETS

**Datasets.** We evaluate CryoNOO on the `IgG-1D` and `IgG-RL` datasets from CryoBench (Jeon et al., 2024), which both simulate the conformational heterogeneity of the human IgG antibody. `IgG-1D` introduces simple conformational variability by simulating a one-dimensional motion, achieved by rotating a single Fab domain of the antibody along a circular arc. In contrast, `IgG-RL` features a more complex ensemble generated by randomly sampling the dihedral angles of the backbone of the flexible linker, resulting in diverse conformations. To generate a dataset of novel views, we simulate the image formation model following the CryoBench (Jeon et al., 2024) pipeline. We sample random poses, CTF defocus values, and additive Gaussian noise to generate a dataset of 100k images. We use a default noise level of SNR 0.1, and additionally evaluate performance under higher noise conditions (SNR 0.01) for the `IgG-1D` dataset. More details can be found in the Appendix.

**Evaluation Metrics.** To evaluate reconstruction quality, we use the per-image Fourier Shell Correlation (FSC), which quantifies similarity between reconstructed volumes and ground truth structures for a heterogeneous dataset. Since ground truth is unavailable for real cryo-EM datasets, this evaluation is limited to synthetic datasets (`IgG-1D`, `IgG-RL`). Following the protocol in Jeon et al. (2024), we randomly sample one image per ground truth structure (100 images total) and reconstruct individual volumes using latent coordinates inferred from the trained encoder. The area under the FSC curve ($FSC_{AUC}$) is computed by comparing each reconstruction to its corresponding ground truth volume.

**Results.** Quantitative results are presented in Table 1. We find that CryoNOO improves generalization without compromising training performance, as training set performance is overall comparable between CryoNOO and CryoDRGN, but CryoNOO significantly improves test performance. The better amortization of CryoNOO is also reflected in the latent space. In Figure 2, we present representative reconstructions of 10 structures each from the training and test sets for `IgG-1D`. These samples are drawn uniformly from the full set of 100 conformations and are not cherry-picked. For `IgG-1D`, CryoNOO successfully captures the 1D circular motion of the Fab region in the latent for both the train and the test set, whereas CryoDRGN only succeeds on the train set. Similarly, CryoNOO successfully reconstructs volumes in both the train and test cases, while CryoDRGN fails on the test set. In the more difficult `IgG-RL` dataset, CryoNOO does a much better job of capturing the orientation of the flexible Fab region in test set reconstructions, while providing better organization in the latent space. However, there still remains a gap between the train and test latent spaces

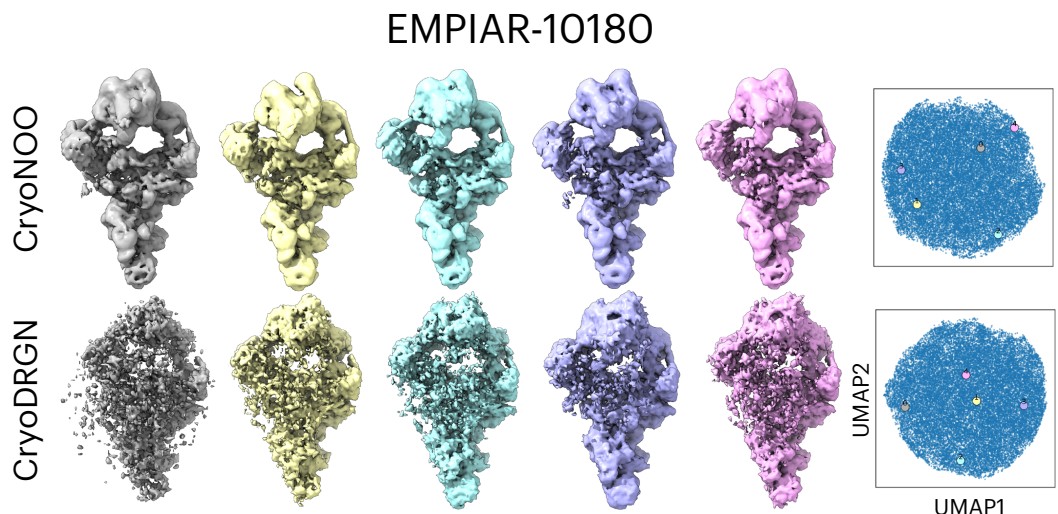

Figure 4: **EMPIAR-10180.** Sampled density maps and latent embeddings visualized using UMAP, colored by K-means cluster labels ($k = 5$). Cluster centroids are indicated, and corresponding density maps illustrate centroid structures. CryoNOO produces much higher resolution reconstruction on the test views than CryoDRGN.

on the `IgG-RL` dataset (Table 1), which we attribute in part to the fact that existing reconstruction methods, such as CryoDRGN, already struggle to reconstruct `IgG-RL` in the overfitting case.

## 4.2 HETEROGENEOUS RECONSTRUCTION OF EXPERIMENTAL DATASETS

**Datasets.** We evaluate our method on two publicly available cryo-EM datasets that exhibit structural heterogeneity. The first is the `SARS-CoV-2 Spike` glycoprotein ectodomain, which displays conformational variability between open and closed receptor-binding domain (RBD) states Walls et al. (2020). We adopt the pre-processing procedure from Levy et al. (2024). The second dataset, `EMPIAR-10180`, contains images of the pre-catalytic spliceosome Plaschka et al. (2017), a highly flexible macromolecular complex. For `EMPIAR-10180`, we follow the preprocessing pipeline described in Zhong et al. (2021). Both datasets are split into training and test sets using an 80/20 ratio. More details are provided in the Appendix.

**Results on SARS-CoV-2 Spike Protein** (Walls et al., 2020). The original study resolved two distinct conformations of the spike protein in closed and open states at high resolution. We apply K-means clustering ($k = 5$) in the latent space and reconstruct density maps from the cluster centroids (Fig. 3). Our method successfully recovers the distinct conformations from the unseen test set, clearly resolving both open and closed RBD states (e.g., yellow and skyblue structures). In contrast, CryoDRGN only reconstructs accurate structures for images in the training set and produces volumes resembling pure noise. Qualitative results on the train data are in Appendix (10).

**Results on EMPIAR-10180** (Plaschka et al., 2017). The inherent flexibility of this complex presents a challenging test case, requiring models to recover a continuous conformational landscape rather than discrete states. In Figure 4, we visualize density maps reconstructed from the centroids of K-means clusters of the test set embeddings. Compared to CryoDRGN, which produces noisy and poorly resolved reconstructions, CryoNOO recovers sharper density maps that better preserve detail. Qualitative results on train data are in Appendix (11).

## 4.3 EVALUATING AMORTIZATION

**Comparison with autodecoding.**

To demonstrate the effectiveness of our approach, we compare against CryoDRGN-AI, an autodecoder method (Levy et al., 2025). Since autodecoding methods have no encoder and do not amortize,

Table 2: **Comparison with autodecoding.** Total Time, Inference Time, Per-image $FSC_{AUC}$, Mean (std), Latent Metrics

| Method | Inference Time | Total Time | $FSC_{AUC}$ ($\uparrow$) | KLD ($\downarrow$) | TV ($\downarrow$) | WD ($\downarrow$) |
|---|---|---|---|---|---|---|
| CryoNOO | 6.04 min | 2:03:42 | **0.367 (0.015)** | **2.319** | **0.2995** | **1.376** |
| CryoDRGN-AI (+ 1ep) | 2.15 min | 3:38:04 | 0.342 (0.002) | 12.471 | 0.762 | 1.609 |
| CryoDRGN-AI (+ 25ep) | 53.75 min | 4:29:40 | 0.365 (0.013) | 12.379 | 0.767 | 1.607 |

Table 3: **Latent Distribution Shift.** We report KLD, TV, JSD, Hellinger (Hell), and Wasserstein distance (WD). Best results per dataset are bolded.

| Dataset | Method | KLD ($\downarrow$) | TV ($\downarrow$) | JSD ($\downarrow$) | Hell ($\downarrow$) | WD ($\downarrow$) |
|---|---|---|---|---|---|---|
| IgG-1D (SNR 0.1) Jeon et al. (2024) | CryoDRGN | 65.8445 | 0.9554 | 0.6319 | 0.9302 | 1.5880 |
| | CryoNOO | **2.3190** | **0.2995** | **0.0725** | **0.2755** | **1.3762** |
| IgG-1D (SNR 0.01) Jeon et al. (2024) | CryoDRGN | 4.1191 | 0.5974 | 0.2657 | 0.5505 | 1.1627 |
| | CryoNOO | **1.3627** | **0.2169** | **0.0372** | **0.1949** | **1.1612** |
| IgG-RL Jeon et al. (2024) | CryoDRGN | 64.2357 | 0.9977 | 0.6875 | 0.9835 | 2.1037 |
| | CryoNOO | **5.9402** | **0.7331** | **0.3791** | **0.6743** | **1.4984** |
| EMPIAR-10180 Plaschka et al. (2017) | CryoDRGN | 9.1861 | 0.7669 | 0.4247 | 0.7234 | 1.9709 |
| | CryoNOO | **1.6079** | **0.5093** | **0.1935** | **0.4614** | **1.0669** |
| Sars-Cov-2-Spike Walls et al. (2020) | CryoDRGN | 9.1861 | 0.7669 | 0.4247 | 0.7234 | 1.9709 |
| | CryoNOO | **2.3950** | **0.6085** | **0.2696** | **0.5529** | **1.1414** |

test-time optimization is required for novel views in order to optimize their latent codes. In contrast, CryoNOO requires no training at inference time and only needs a single forward pass of the network.

Specifically, starting with the latent codes from a CryoDRGN pre-trained for 50 epochs, we first train CryoDRGN-AI on the training data. At inference time, we continue training on the test data until reaching a comparable per-image FSC score. As shown in Table 2, our method generalizes to novel views without any additional training (via amortization), whereas CryoDRGN-AI requires about 25 extra epochs. Since CryoNOO only requires one forward pass for inference, CryoNOO achieves a nearly 95% efficiency gain over the autodecoder baseline. Moreover, in terms of the latent distribution comparison, our approach outperforms CryoDRGN-AI. Importantly, CryoDRGN-AI also requires initialization from the latent codes of a CryoDRGN pre-trained for 50 epochs and 50 epochs of additional training in addition to the 25 updates during the autodecoding inference process, so the complete pipeline requires more than twice the total time of CryoNOO. Total time is presented in Table 2.

**Train-Test Overlap in Latent Space.** Table 3 presents the KL divergence, Total Variation, JSD, Hellinger distance, and Wasserstein distance between training and test latent distributions for CryoDRGN and our method across five datasets, including both synthetic (IgG-1D, IgG-RL) and real (EMPIAR-10180, SARS-CoV-2 Spike) cases. Across all datasets, our method consistently achieves significantly lower latent distance compared to CryoDRGN.

Figure 2 shows a PCA visualization of the learned latent space for both our method (top) and CryoDRGN (bottom) on the IgG-1D dataset. We project the training and test latent embeddings onto the first two principal components computed from the training set. Our method yields a well-aligned and continuous latent manifold across both training and test samples. The test embeddings (top middle) are overlapped with the same principal subspace structure as the training embeddings (top right), and their overlay (top left) shows a high level of overlap, indicating that CryoNOO has successfully amortized.

In contrast, CryoDRGN exhibits poor alignment between training and test embeddings. While the training set displays clear continuous variation (bottom right), the test set (bottom middle) collapses into a highly concentrated region, and the overlay (bottom left) shows a severe mismatch in latent

coverage. These observations suggest that CryoDRGN overfits to seen views and fails to extend its latent representation meaningfully to novel inputs.

## 4.4 ABLATION STUDY

We conducted an ablation study on the `IgG-1D (SNR 0.1)` dataset to evaluate the contribution of each component in our framework. Unless otherwise stated, we follow CryoDRGN defaults, training for 50 epochs with a batch size of 64 and an expander dimension of 1024. Our method is activated after $N = 10$ epochs of standard CryoDRGN training, with noise factors of 1, 2, and 3, and a VICReg loss weight of $\lambda = 0.3$.

We found that skipping the CryoDRGN warm-up phase led to degraded performance, while activating our method after 5–10 epochs yielded the best results, highlighting the importance of early-stage reconstructions for generating meaningful denoised views. A single noise-augmented view proved sufficient for strong generalization, and we observed that both over-regularization ($\lambda = 1$) and under-regularization ($\lambda = 0.1$) impaired performance. Although VICReg typically uses high-dimensional expanders (e.g., 8192), our method performs well with significantly lower dimensional expanders. Using no expander performs similarly to using an expander with poor dimension. Finally, we tested SimSiam (Chen & He, 2021) as an alternative self-supervised framework, but found that the performance of SimSiam is worse than that of VICReg, showing that improvements in self-supervised learning methods can directly translate to our setting. Using only an invariance loss is even worse, showing the need to

Table 4: **Ablation Study on IgG-1D (SNR 0.1).** Per-image $\text{FSC}_{\text{AUC}}$. Mean (std).

| Variant | $\text{FSC}_{\text{AUC}}$ |
|---|---|
| **Ours** | **0.367** (**0.015**) |
| w/o CryoDRGN warmup | 0.259 (0.035) |
| + warmup ($N = 5$) | 0.363 (0.015) |
| + warmup ($N = 40$) | 0.34 (0.019) |
| Single noise | 0.364 (0.013) |
| $\lambda = 1.0$ | 0.349 (0.022) |
| $\lambda = 0.1$ | 0.345 (0.02) |
| **Expander dim = 2048** | **0.367** (**0.014**) |
| Expander dim = 32 | 0.310 (0.034) |
| Expander dim = 128 | 0.361 (0.018) |
| Expander dim = 4096 | 0.366 (0.015) |
| No Expander | 0.345 (0.024) |
| Only Invariance | 0.331 (0.012) |
| SimSiam Chen & He (2021) | 0.344 (0.016) |

use self-supervised learning to produce noise-invariant representations. Additional details are provided in the Appendix.

## 5 DISCUSSION

We introduce the first fully amortized inference framework for generalizable cryo-EM reconstruction under novel views. Leveraging the inherent denoising capability of CryoDRGN, we develop a self-distillation approach using noise-augmented views to learn noise-invariant representations. Where existing methods fail to reconstruct 3D structures from novel test distributions, our approach succeeds, preserving stable and consistent latent-space distributions across all evaluated datasets. Thanks to its fully amortized inference design, the framework scales effectively to large datasets with minimal computational overhead. Our method is promising technique for reducing the computational cost of cryo-EM reconstruction, allowing the user to potentially train on only a small subset of a large cryo-EM imaging dataset but perform inference on the whole dataset. Additionally, improving the generalizability of cryo-EM reconstruction models could enable high-quality initial models to be trained from much smaller subsets of particles, which could then bootstrap traditional refinement workflows, drastically reducing computational costs.

CryoNOO represents a step toward cryo-EM models that do not overfit to the training set. While our method focuses on generalizing to unseen views of the same protein, generalizing to new protein structures is an exciting frontier challenge for cryo-EM reconstruction. We envision a future where accurate structures can be reconstructed from relatively few observations, without requiring millions of identical particles. Achieving this vision will require models that learn strong priors over protein images and 3D structures from cryo-EM data and generalize robustly beyond the training distribution. CryoNOO takes a first step in this direction by providing a simple yet effective strategy for improving generalization under the high-noise conditions intrinsic to cryo-EM.

**Ethics statement.** Our method for generalizable cryo-EM reconstruction under novel views helps increase the understanding of the biological functions of the reconstructed biomolecules. We do not believe that our method has any negative societal impacts. LLM usage was limited to improving the writing of the paper.

**Reproducibility statement.** We will release the code and dataset upon publication.

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

## A BENCHMARK DATASETS

### A.1 SYNTHETIC DATASETS

**IgG-1D.** IgG-1D is a simulated, diagnostic conformational heterogeneity dataset from CryoBench Jeon et al. (2024) derived from the atomic model of human IgG antibody. Heterogeneity was introduced by rotating a single Fab domain about its hinge in a circular arc. The dataset consists of 100,000 synthetic images (128×128 pixels) at a pixel size of 3 Å/pix.

**IgG-RL.** IgG-RL is a more challenging simulated conformational ensemble dataset from CryoBench Jeon et al. (2024) based on the same IgG antibody, but with a flexible linker region sampling a rich set of configurations. Instead of a single rotation, the heterogeneity here is generated by randomizing the backbone dihedral angles of the antibody's hinge according to statistical distributions of disordered peptides, which produces a broad range of IgG conformations not confined to a 1D trajectory, including bends and twists of the Fab arms. Like IgG-1D, 100,000 projections were simulated at 128² pixels (3 Å/pix).

**Novel View Datasets.** For the novel view datasets, we follow the dataset generation protocol described in Jeon et al. (2024). Starting from $256 \times 256 \times 256$ density maps provided by CryoBench Jeon et al. (2024), we generate 2D projections by uniformly sampling poses $R \in SO(3)$ and in-plane translations $t \in [20, 20]^2$ pixels. The contrast transfer function (CTF) is modeled with an accelerating voltage of 300 kV, a spherical aberration at 2.7 mm, and an amplitude contrast of 0.1. Defocus values are randomly sampled without replacement from EMPIAR-11247 Feathers et al. (2022). Gaussian noise is added to achieve a signal-to-noise (SNR) ratio of 0.1 and 0.01. We simulate 1,000 images per conformation, resulting in a total of 100,000 images. Finally, the images are downsampled to $D = 128$ using Fourier cropping.

### A.2 EXPERIMENTAL DATASETS

**EMPIAR-10180.** This is an experimental cryo-EM dataset of the pre-catalytic spliceosome (yeast B complex) featuring large-scale continuous structural rearrangements Zhong et al. (2021). The dataset contains roughly 327,490 images of the spliceosomal complex with 320×320 pixels and pixel size 1.7 Å/pix. Our experiments use a filtered subset of these particles focusing on intact spliceosome images. The high flexibility of this complex provides a challenging test on experimental data as methods must recover a continuous bending/twisting landscape rather than a few discrete states. The dataset is split into training and test sets with an 80/20 ratio, resulting in 124,197 training samples and 31,050 test samples.

**SARS-CoV-2 Spike.** This is a single-particle experimental dataset of the SARS-CoV-2 spike glycoprotein ectodomain exhibiting primarily conformational heterogeneity between open and closed receptor-binding domain configuration Walls et al. (2020). This dataset comprises roughly 370,000 particle images of the spike ectodomain, which was downsampled to 128×128 pixels at 3.3 Å/pix for efficiency. The dataset is split into training and test sets with an 80/20 ratio, resulting in 165,008 training samples and 41,252 test samples.

## B IMPLEMENTATION DETAILS

**Optimization.** We train all models in Pytorch using 1 NVIDIA A100 GPU. For the synthetic datasets (IgG-1D and IgG-RL), we follow CryoDRGN Zhong et al. (2019; 2021) using the same training schedule and optimization parameters. We use Adam Kingma (2014) optimizer with learning rate set to 0.0001 without weight decay. Moreover, we applied learning rate scheduling using `torch.optim.lr_scheduler.ReduceLROnPlateau`, which reduces the learning rate when the total loss stops decreasing. Specifically, we used the scheduler in `min` mode with a reduction factor of 0.5 and a patience of 3 epochs. This scheduling strategy was only applied to CryoNOO, as no improvement was observed when applied to CryoDRGN Zhong et al. (2019) (0.331 vs 0.330 FSC_AUC).

**Noise estimation for experimental datasets.** Following the cryo-EM image formation model (Eqn. 6), we assume that the noise in cryo-EM images is additive mean zero Gaussian noise $\epsilon \sim \mathcal{N}(0, \sigma)$

in real space. Thus, to create noise-augmented images that resemble the data distribution from denoised images, it suffices to estimate $\sigma$ of the Gaussian additive noise.

Our strategy for estimating $\sigma$ consists of masking the signal and estimating $\sigma$ as the standard deviation of the background pixels. We derive our noise estimation as follows: first, we note that almost all of the power of a cryo-EM image is concentrated at the low frequencies in the power spectrum. Thus, the high frequencies of the power spectrum are dominated by the power spectrum of Gaussian noise, which has a constant power spectrum $\sigma^2$. To compute $\sigma$, we calculate the power spectrum of the image and average the power at all frequencies $k$ bigger than some threshold $k_{\text{thresh}}$. We can then calculate $\sigma$ using the Plancherel theorem for the 2D discrete Fourier transform as $\sigma = \sqrt{\frac{P}{D^2}}$. The complete algorithm is given in Algorithm 1. The `radial_average` function takes a 2D array and averages the values of the ray over concentric rings around the center of the 2D array.

---

**Algorithm 1** Noise estimation from cryo-EM images

---

**Require:** Cryo-EM image $I_n \in \mathbb{R}^{D \times D}, n = 1, \ldots, N$, frequency threshold $k_{\text{thresh}}$
1:   $\sigma\text{s} \leftarrow []$
2: **for** $j \leftarrow 1, \ldots, N$ **do**
3:     $\hat{I}_j \leftarrow \text{FFT}(I_j)$
4:     $PS \leftarrow |\hat{I}_j|^2$
5:     $rad \leftarrow \texttt{radial\_average}(PS)$
6:     $P \leftarrow \frac{1}{D - k_{\text{thresh}}} \sum_{k=k_{\text{thresh}}}^{D} r[k]$
7:     $\sigma\text{s} \leftarrow \sigma\text{s} \cup \{\sqrt{P/D^2}\}$
8: **end for**
9: **return** $\sigma = \frac{1}{N} \sum_{i=1}^{N} \sigma\text{s}[i]$

---

**Noise factors ($f_j$).** We multiply each noise factor $f_j$ with the base standard deviation $\sigma$ to generate Gaussian noise, i.e., $\epsilon \sim N(0, f_j\sigma)$. For synthetic datasets, we can reproduce the exact noise level by simulating images using the image formation model described in Jeon et al. (2024), starting from the `before_noise` images. Since the original noise level ($\sigma$) was estimated from $256 \times 256$ images and we downsampled them to $128 \times 128$, the effective noise level increases by a factor of 2. Therefore, applying a *noise factor* of 2 restores the original noise level. For experimental datasets, we use Algorithm 1 to estimate the noise level $\sigma$, and perform a grid search to determine appropriate noise factors. The detailed noise factors ($f_j$) and noise levels ($\sigma$)—including the original $\sigma$ for synthetic datasets and the estimated $\sigma$ for experimental datasets—are shown in Table 5.

Table 5: **Noise standard deviation (sigma) for each dataset.**

| Dataset | Original / Estimated $\sigma$ | Noise factors $f_j$ |
|---|---|---|
| IgG-1D (SNR 0.1) | 1.6979 | 1, 2, 3 |
| IgG-1D (SNR 0.01) | 5.3662 | 1.5, 2, 2.5 |
| IgG-RL (SNR 0.1) | 1.6833 | 1.5, 2, 2.5 |
| EMPIAR-10180 | 1.5396 | 9, 10, 11 |
| SARS-CoV-2 Spike | 3.2813 | 9, 10, 11 |

**Architecture.** We adopt the standard CryoDRGN Zhong et al. (2019) architecture, consisting of a VAE with fully connected encoder and decoder networks. Both the encoder and decoder are 3-layer residual MLPs with 1024 hidden dimensions, and the latent variable $z$ is 8-dimensional. For the expander $h$, we use a 3-layer MLP with 1024 hidden dimensions per layer for the IgG-1D (SNR 0.1) and IgG-RL datasets, and 2048 hidden dimensions for the remaining datasets. BatchNorm and ReLU activations are applied after each layer, except the last layer.

**Masking.** In the original CryoDRGN setting, a circular spatial mask is applied during both decoding and loss computation for computational efficiency and improved performance, while the encoder processes the full image. In contrast, in our setting, the mask is applied only during loss computation, as we retain the full image size in order to add noise after the inverse Fourier transform.

Table 6: **Computational costs** We report reconstruction quality (per-image FSC on novel views), total model memory, and training time for IgG-1D (SNR 0.1).

| Model | FSC_AUC (Novel Views) | Memory (MB) | Time (hh:mm:ss) |
|---|---|---|---|
| CryoDRGN | 0.330 (0.012) | 75.84 | 1:42:17 (2:02/epoch) |
| CryoNOO (1 noise) | 0.364 (0.013) | 83.97 | 1:49:27 (2:44/epoch) |
| CryoNOO (3 noises) | 0.367 (0.015) | 83.97 | 1:57:38 (2:56/epoch) |

Table 7: **Additional experiments.** IgG-1D (SNR 0.1).

| Model | FSC_AUC (Novel Views) |
|---|---|
| CryoDRGN | 0.330 (0.012) |
| CryoDRGN + Crop / Flip / Rotations | 0.334 (0.008) |
| CryoDRGN + Noise Aug | 0.333 (0.011) |
| CryoDRGN | 0.330 (0.012) |
| CryoNOO | 0.367 (0.015) |
| Decoder freeze | 0.359 (0.009) |
| Noise factors ($\times 2$) | 0.361 (0.019) |
| Noise factors ($\div 2$) | 0.358 (0.018) |

## C  COMPUTATIONAL COSTS

Table 6 compares CryoDRGN Zhong et al. (2019) and our proposed CryoNOO in terms of reconstruction accuracy, memory usage, and training time. Specifically, we report FSC_AUC on novel views for the IgG-1D (SNR 0.1) dataset, along with the total model memory, and total training time over 50 epochs with a batch size of 64. Although a direct comparison is not entirely fair—since CryoDRGN is not fully amortized—we compare computational cost and generalization performance. The performance gap between using a single noise factor and three noise factors in CryoNOO is relatively small, so we include results for both settings.

## D  ADDITIONAL EXPERIMENTS

**Conventional data augmentations.** Conventional augmentations may partially mitigate overfitting, but does not necessarily enforce invariance to different noise realizations in the latent space. Our approach explicitly trains for this invariance via denoising. We tested traditional real-space augmentations (cropping, rotations, and flips) and found that they did not improve novel-view generalization—the AUC-FSC on the IgG-1D (SNR 0.1) test set was 0.334±0.008, comparable to the cryoDRGN baseline. Similarly, adding noise yielded no improvement, with an AUC-FSC of 0.333±0.011 (Table 7).

**Decoder freeze.** Instead of using the self-distillation setup, we freeze the decoder in CryoNOO. After a warm-up phase with standard CryoDRGN training, the decoder weights are fixed. This results in lower performance compared to our default self-distillation setting (Table 7).

**Different noise factors.** To evaluate the sensitivity of our method to the choice of noise factors ($f_j$), we experiment with different settings. Our default uses noise factors of 1, 2, and 3. We additionally test with scaled variants: 2, 4, 6 ($\times 2$) and 0.5, 1, 1.5 ($\div 2$). As shown in Table 7, although performance decreases with these alternative noise settings, the FSC_AUC remains substantially higher than that of CryoDRGN Zhong et al. (2019).

**Kernel width in Kde.** In Table 3, we present the alignment between the train and test latent distributions using K-Fold cross-validation, where the bandwidth search space ranges from 0.1 to 10 in intervals of 0.2. The log-likelihood of the data is optimized during cross-validation. The selected bandwidths are reported in Table 8.

Table 8: **Bandwidths used in distribution shift experiments.**

| Dataset | Method | Bandwidth |
|---|---|---|
| IgG-1D (SNR 0.1) | CryoDRGN | 0.3 |
| | CryoNOO | 0.7 |
| IgG-1D (SNR 0.01) | CryoDRGN | 0.7 |
| | CryoNOO | 0.7 |
| IgG-RL | CryoDRGN | 0.7 |
| | CryoNOO | 0.9 |
| EMPIAR-10180 | CryoDRGN | 0.9 |
| | CryoNOO | 0.7 |
| Sars-Cov-2-Spike | CryoDRGN | 0.9 |
| | CryoNOO | 0.7 |

Table 9: **Distribution metrics** Comparison of CryoDRGN and CryoNOO across different bandwidths on IgG-1D (SNR 0.1). Best results per row are bolded.

| Method | Bandwidth | KLD | TV | JSD | Hellinger | WD |
|---|---|---|---|---|---|---|
| CryoDRGN | 0.1 | 613.8291 | 0.9996 | **0.6923** | 0.9964 | 1.5880 |
| CryoNOO | 0.1 | **76.1661** | **0.9994** | 0.6921 | **0.9956** | **1.3762** |
| CryoDRGN | 0.5 | 28.8464 | 0.8739 | **0.5382** | 0.8422 | 1.5880 |
| CryoNOO | 0.5 | **4.3269** | **0.4828** | 1.7720 | **0.4425** | **1.3762** |
| CryoDRGN | 1.0 | 11.5941 | 0.8045 | 0.4492 | 0.7477 | 1.5880 |
| CryoNOO | 1.0 | **1.4308** | **0.2030** | **0.0342** | **0.1868** | **1.3762** |
| CryoDRGN | 3.0 | 0.7760 | 0.3591 | 0.0927 | 0.3103 | 1.5880 |
| CryoNOO | 3.0 | **0.3320** | **0.0583** | **0.0025** | **0.0503** | **1.3762** |

To provide an additional comparison for Table 3, we present results comparing CryoNOO and CryoDRGN trained on IgG-1D (SNR 0.1) on distribution matching metrics at various settings of the kernel width. We find that CryoNOO shows better performance over the different bandwidths. we compute the latent metrics with bandwidth 0.1, 0.5, 1, and 3.

## E    CRYO-EM IMAGE FORMATION MODEL

The goal of cryo-electron microscopy (cryo-EM) reconstruction is to recover a 3D molecular structure $V : \mathbb{R}^3 \to \mathbb{R}$ from a collection of noisy 2D projection images $X_1, \ldots, X_N$. Each projection corresponds to an individual particle captured in an unknown pose, parameterized by a rotation $R \in SO(3)$ and an in-plane translation $t \in \mathbb{R}^2$. The image formation process for each observation $X_i$ can be modeled in real space as:

$$X(r_x, r_y) = g * \left( \int_{\mathbb{R}} V(R^\top r) \, dr_z + t \right) + \epsilon, \tag{6}$$

where $r = (r_x, r_y, r_z)^\top$, $t = (t_x, t_y)$ is the translation vector, and $\epsilon$ denotes additive noise introduced during imaging. The function $g : \mathbb{R}^2 \to \mathbb{R}$ represents the microscope's point spread function (PSF), which is convolved with the 2D projection of the 3D volume. In practice, reconstruction is generally performed in Fourier space by leveraging the Fourier Slice Theorem (FST) Bracewell (1956), which simplifies the computation of the image formation model by replacing the projection integral with taking a slice through the 3D Fourier transform of the volume.

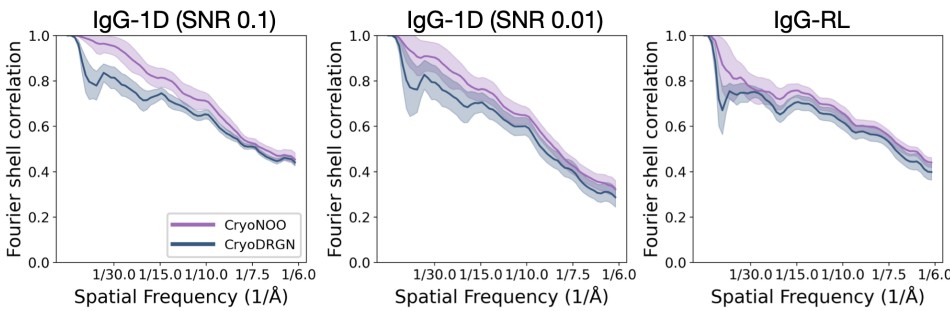

Figure 5: **Per-Image FSC.** Each curve shows the average FSC curve across all conformations with error bars indicating the stan- dard deviation. Colors correspond to methods shown in the legend.

## F  PER-IMAGE FSC FOR HETEROGENEOUS RECONSTRUCTION

Following Jeon et al. (2024), we use per-image FSC to jointly evaluate reconstruction quality and structural heterogeneity. In cryo-EM, the Fourier Shell Correlation (FSC) curve is a standard tool for comparing two volumes by measuring their correlation across concentric shells in the Fourier domain. Since cryo-EM methods often reconstruct a 3D volume from a single 2D image, per-image FSC quantifies how well each individual reconstruction matches the ground truth conformation. Specifically, we compute the FSC between the reconstructed volume from a single image and its corresponding ground truth, then summarize performance using the area under the FSC curve (FSC_AUC). To capture the distribution of reconstruction quality across conformations, we sample one image per conformation as in Jeon et al. (2024). The maximum value of FSC_AUC is 0.5.

The average per-image FSC curves for CryoNOO and CryoDRGN on the IgG-1D (SNR 0.1 and SNR 0.01) and IgG-RL datasets are shown in Figure 5, with corresponding FSC_AUC values summarized in Table 1.

## G  QUALITATIVE RESULTS

In this section, we present additional examples of reconstructed volumes from both CryoNOO and CryoDRGN across the test view datasets. Figures 6, 7, and 8 compare CryoNOO and CryoDRGN Zhong et al. (2019) in 10 representative structures of the IgG-1D (SNR 0.1 and SNR 0.01) and IgG-RL datasets, respectively. For the IgG-1D (SNR 0.1) dataset, we also include results from SimSiam Chen & He (2021). These examples correspond to the structures shown in Figure 2, and are uniformly sampled from the full set of 100 conformations without cherry-picking.

Figure 6 shows reconstructions of 10 representative IgG-1D (SNR 0.1) structures by CryoNOO, CryoDRGN Zhong et al. (2019), and SimSiam Chen & He (2021), compared to the ground truth (G.T). CryoNOO consistently reconstructs the correct rotation of the right Fab domain. In contrast, CryoDRGN often fails to recover meaningful density in the flexible Fab region, resulting in missing structures. SimSiam performs better than CryoDRGN in recovering the flexible region, but still fails to capture the correct rotation, as highlighted in the boxed regions. Other unboxed examples also exhibit incorrect conformations.

Figure 7 demonstrates that under more challenging noise conditions, CryoDRGN fails to recover the correct rotation direction entirely. None of the structures are accurately reconstructed, and the per-image FSC scores primarily reflect fixed regions rather than the flexible parts.

For the more challenging IgG-RL dataset, CryoDRGN fails to reconstruct even a plausible "Fab-like" shape (Fig. 8 and Fig. 9). As in the IgG-1D (SNR 0.01) dataset, it does not capture meaningful density in the flexible regions. In contrast, CryoNOO successfully recovers these regions and reconstructs them with the correct orientation.

Finally, we include qualitative results on the SARS-CoV-2 and EMPIAR-10180 training data in Figure 10, 11.

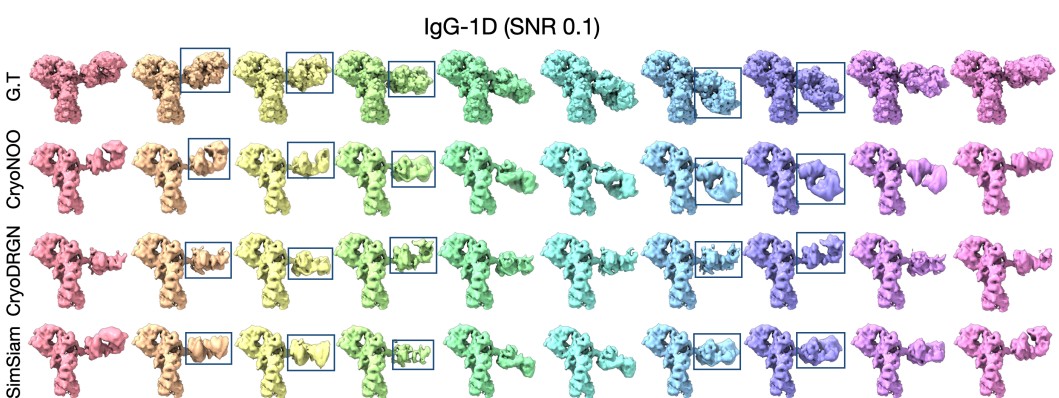

Figure 6: **IgG-1D (SNR 0.1).**

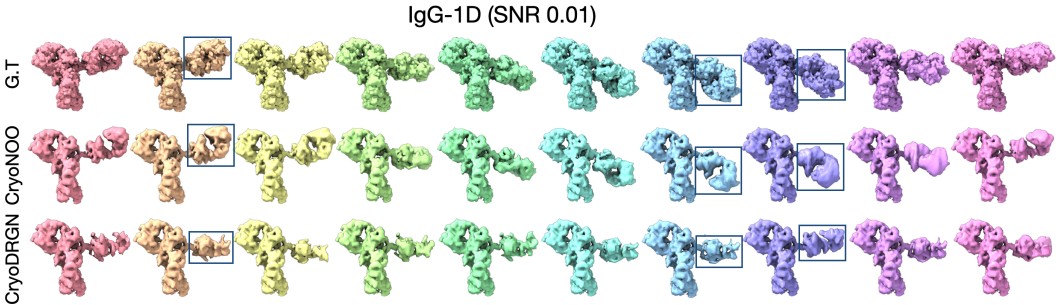

Figure 7: **IgG-1D (SNR 0.01).**

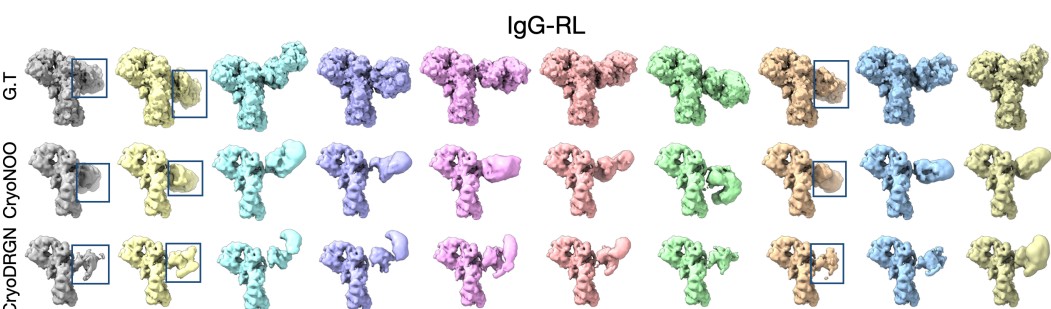

Figure 8: **IgG-RL.**

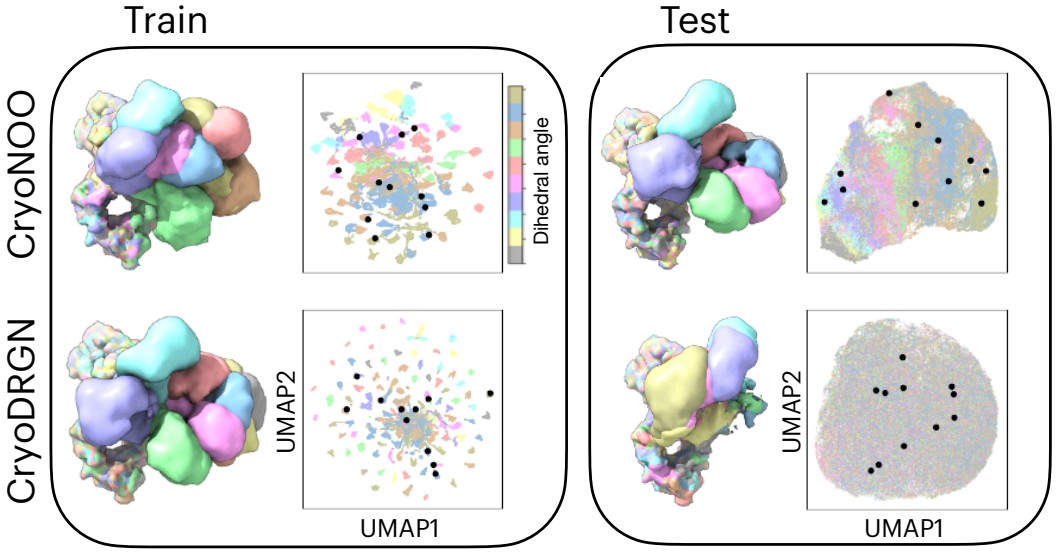

Figure 9: **IgG-RL.** CryoNOO can generally recover the orientation of the protein's arm and appears to have a better organization in latent space, with different conformations beginning to cluster together, whereas we do not see any evidence of clustering in the latent space for CryoDRGN.

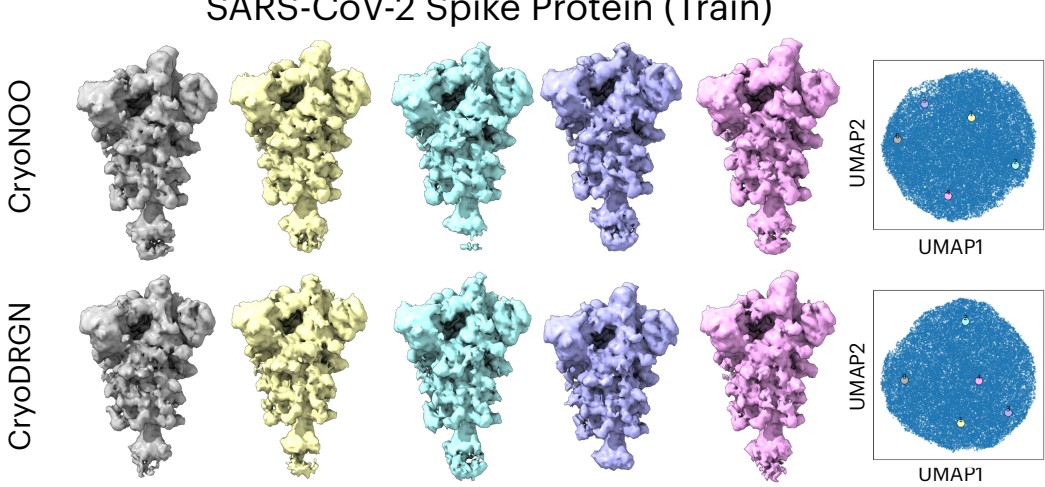

Figure 10: **SARS-CoV-2 Spike Protein (Train).**

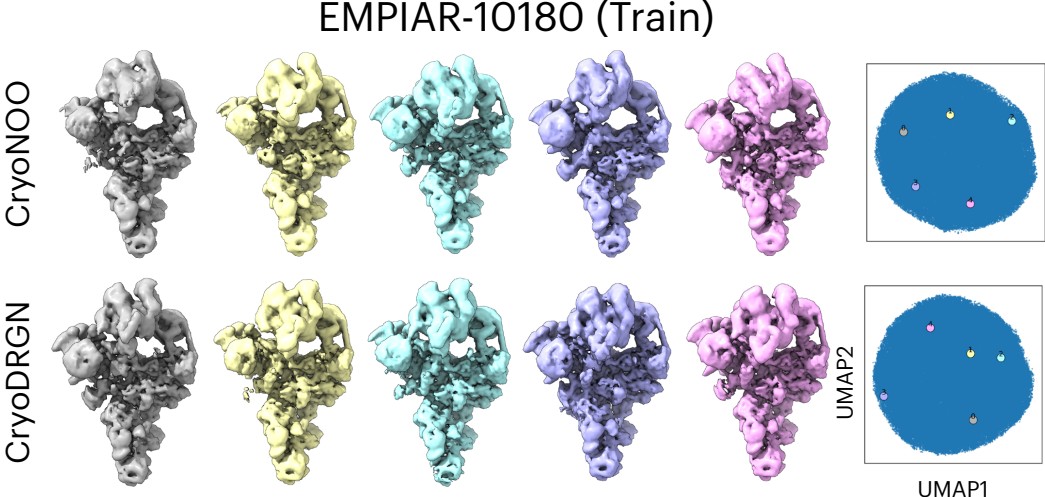

Figure 11: **EMPIAR-10180 (Train).**

