# OpenReview forum: "Separating signal from noise: a self-distillation approach for amortized heterogeneous cryo-EM reconstruction"
_ICLR.cc/2026/Conference — ICLR 2026 Conference Withdrawn Submission_

### Official Review · Reviewer_JbPZ · 2025-10-27

**Soundness:** 3
**Presentation:** 3
**Contribution:** 2
**Rating:** 4
**Confidence:** 4

**Summary:**

This paper introduces CryoNOO, a self-distillation framework designed to improve the generalization of image encoders in cryo-EM neural reconstruction methods (e.g., cryoDRGN) to novel views. The approach is motivated by the insight that cryo-EM reconstruction can be interpreted as a multi-view denoising problem. Once pre-trained, cryoDRGN functions as an implicit teacher (denoiser). Given a real particle image, synthetic noise is injected into its denoised output to create augmented views of the same orientation. These augmented images are processed by the same encoder, producing a latent distribution. To align the latent representations of real and synthetic images, the framework employs a VICReg-based loss, encouraging the extraction of view-invariant features and enhancing generalization. Experiments on synthetic and experimental datasets show that CryoNOO enables robust generalization to unseen images corresponding to the same underlying 3D structure.

**Strengths:**

1. Originality: This work introduces a novel self-distillation strategy to enhance the generalization capability of image encoders in cryo-EM neural reconstruction. To the best of my knowledge, it is the first to leverage a pre-trained cryoDRGN model as an implicit teacher within a self-supervised framework using a VICReg-based objective.
2. Quality of Results: The proposed method, CryoNOO, yields clear improvements in the structure of the latent space on unseen test data. Experiments on both synthetic and real datasets demonstrate that the encoder effectively generalizes to previously unseen particle images corresponding to the same underlying 3D structure, the results seems to be promising and reproducible.

**Weaknesses:**

1. Motivation and Practical Relevance: The motivation for improving encoder generalization to unseen views is not clearly articulated and appears somewhat counterintuitive in the context of cryo-EM. In typical heterogeneous reconstruction workflows, all available particle images are used during training to maximize reconstruction quality. The primary goal in cryo-EM is to improve the resolution of the reconstructed structure; when resolution improvements plateau, a secondary goal is to increase reconstruction throughput. In the context of AI-based methods, both objectives are closely tied to the generalization capacity of a universal model.

    While the paper demonstrates that training on 80% of the data and testing on 20% unseen data leads to improved encoder performance on held-out images, this result is not directly linked to final 3D reconstruction quality (e.g., resolution, structural continuity) or reconstruction efficiency. In fact, the introduction of additional network components and loss terms could potentially increase training complexity and slow convergence, but this aspect is not experimentally evaluated. A promising direction would be to examine whether improved generalization allows for latent space interpolation—rather than searching for nearest neighbors in the input space—which could indicate that CryoNOO learns a more meaningful latent representation beyond merely fitting noisy image-to-structure mappings.

1. Methodological Justification: It remains unclear why injecting simple synthetic noise leads to improved generalization on real data. This choice is insufficiently motivated, especially given that it may not reflect realistic imaging conditions. A more comprehensive evaluation—including standard image-level augmentations (e.g., rotation, flipping) and comparisons to training on the full dataset—would clarify the unique benefits of the proposed approach. Moreover, it would be valuable to discuss whether the method can be extended toward a more universal pre-training framework, for example by training across multiple datasets to enhance reconstruction throughput. Such an investigation could strengthen the practical significance of CryoNOO beyond its current experimental scope.

**Questions:**

1. Could the authors better justify the practical motivation of improving encoder generalization in cryo-EM, and its concrete impact on reconstruction resolution or throughput?
2. How does the proposed method affect final 3D reconstruction performance (e.g., resolution, pose estimation accuracy, efficiency) compared to cryoDRGN trained on the full dataset?
3. Could the authors clarify the rationale and advantage of synthetic noise over standard augmentations and provide supporting evidence?
4. Will the code and model be released, and what are the potential applications of CryoNOO in discovering new structures or accelerating reconstruction workflows?

---

### Official Review · Reviewer_qMTk · 2025-11-01

**Soundness:** 2
**Presentation:** 2
**Contribution:** 2
**Rating:** 4
**Confidence:** 4

**Summary:**

The paper presents Cryo-NO-Overfit (CryoNOO), an autoencoder-based method for cryo-EM heterogeneous reconstruction. Building upon CryoDRGN, the method integrates two main improvements: 1) a noise-augmented training strategy that distills knowledge from CryoDGRN model and 2) a VICReg-based regularization that enforces consistency and stability in the latent space. Experiments on synthetic and real data  demonstrate that CryoNOO achieves improved generalization and reconstruction quality compared to the CryoDRGN baseline.

**Strengths:**

- The paper presents an extension of CryoDRGN to improve its generalization capability through noise-augmented training and latent space regularization.

- The ablation study clearly demonstrates the individual contributions of each proposed component.

**Weaknesses:**

- Limited novelty. The proposed CryoNOO framework largely builds upon the existing VAE-based CryoDRGN architecture, incorporating two well-known techniques—data augmentation and VICReg regularization. While these additions are sensible, their conceptual contribution is incremental. Moreover, noise-based data augmentation has previously been explored in the cryo-EM domain (e.g., Cryo-GEM, NeurIPS 2024) to improve generalization under similar motivations.

- Technical limitations for heterogeneous reconstruction. The proposed self-distillation mechanism depends on the reconstruction quality of CryoDRGN, which constrains the potential performance gains. Both the noise augmentation and latent regularization components are general training strategies, not specifically designed to address the challenges of heterogeneous 3D reconstruction.

- Insufficient experimental validation. The evaluation primarily compares to CryoDRGN(-AI), lacking broader comparisons with other recent generative cryo-EM methods such as CycleDiffusion, CycleNet, and CryoGEM. The reported quantitative improvements over CryoDRGN-AI are marginal (Table 2), and qualitative results (Figures 3–4) suggest that CryoNOO produces reconstructions with lower spatial resolution than the baseline. Furthermore, the claim of reconstructing “novel particles unseen during training” is not empirically supported by dedicated experiments or clear metrics.

**Questions:**

- How does the proposed noise-augmentation strategy compare to other data augmentation techniques commonly used in cryo-EM or generative modeling?

- Can the authors provide comparative results with other recent generative cryo-EM approaches (e.g., CycleDiffusion, CycleNet, CryoGEM)?

- Could the authors clarify and substantiate the claim that this is the first demonstration of amortized heterogeneous 3D reconstruction of unseen particles? What specific evidence supports this statement?

---

### Official Review · Reviewer_QY4b · 2025-11-01

**Soundness:** 1
**Presentation:** 2
**Contribution:** 1
**Rating:** 2
**Confidence:** 4

**Summary:**

The authors propose CryoNOO, an extension to VAE-based cryo-EM reconstruction methods that enable them with the capability of further processing new images of the same protein density after training.

**Strengths:**

- The paper is easy to follow.
- The usage of VISReg is relatively novel.

**Weaknesses:**

- The setting of generalization to new particle images is misleading. For cryoDRGN, its latent space after training can be interpolated and traversed, and the reconstruction from these interpolated latents follow their meaning, for example the rotation angle in IgG-1D dataset.
- The authors seem do not understand the heterogeneous cryo-EM reconstruction setting. For example, after the cryoDRGN training on a dataset, the next step is to examine the reconstructed latent space by using different latents to perform the reconstruction, without the need of any novel angle/CTF/noise images to be used as the input.
- The dataset creation (Line 300-303) do not respect common cryo-EM practice. In real-world experiments, the CTF value is always fixed per dataset, but the authors also change this value to create “new” images, which apparently deviates from common practice and could lead to the degradation of cryoDRGN reconstructions.

**Questions:**

- Please further explain why you think it is necessary to have “new images” be used as the input to observe their reconstruction, rather than directly mixing them together with other training images?
- Why you think it is insufficient to interpolate and traverse the latent space from training?

---

### Official Review · Reviewer_HoYY · 2025-11-03

**Soundness:** 3
**Presentation:** 3
**Contribution:** 2
**Rating:** 4
**Confidence:** 5

**Summary:**

This paper proposed a simple yet effective strategy to generalize to images not in the training set: learning noise invariant representations.  Impressively, extensive empirical evaluations on both synthetic and experimental datasets demonstrate that this method dramatically improves reconstruction quality on unseen test data

**Strengths:**

1. This paper introduced a self-distillation technique that leverages the encoder-based reconstruction method itself to generate augmented views of training samples and use self-supervised learning to enforce invariance.

2. The methods is very simple and well adapted from the original ViCReg for cryo-EM reconstruction.

3. Various benchmark demonstrated the methods' superior performances.

**Weaknesses:**

1. The generalizability of this method is not validated. Can we only use CryoDRGN as the teacher to do self-distillation? I think other additional methods should be used as teachers to validate the methods' generalizability.

2. Lack of common baseline evaluation:
2.1 The performances on the basic homogeneous setting is not validated.
2.2 The performances on challenging setting: compostional setting is not tested.

3. Lack comparisons with current SOTA approaches, such as CryoNeRF, CryoFM. The performances relative to those methods are not validated.

4. The loss balancement is not carefully invested. The three loss terms may have different values to achieve better and stable performances.

**Questions:**

Listed in weakness

---

### Note · Authors · 2025-11-12

I have read and agree with the venue's withdrawal policy on behalf of myself and my co-authors.